

# Construction and validation of prognosis and treatment outcome models based on plasma membrane tension characteristics in bladder cancer

Zhipeng Wang[1,2,*], Sheng Li[1,2,*], Fuchun Zheng[1,2], Situ Xiong[1,2], Lei Zhang[1,2], Liangwei Wan[1,2], Chen Wang[1,2], Xiaoqiang Liu[1,2] and Jun Deng[1,2]

[1] Department of Urology, First Affiliated Hospital of Nanchang University, Nanchang, Jiangxi, China
[2] Jiangxi Institute of Urology, Nanchang, Jiangxi, China
[*] These authors contributed equally to this work.

## ABSTRACT

**Background**. Plasma membrane tension-related genes (MTRGs) are known to play a crucial role in tumor progression by influencing cell migration and adhesion. However, their specific mechanisms in bladder cancer (BLCA) remain unclear.

**Methods**. Transcriptomic, clinical and mutation data from BLCA patients were collected from The Cancer Genome Atlas (TCGA) and Gene Expression Omnibus (GEO) databases. Clusters associated with MTRGs were identified by consensus unsupervised cluster analysis. The genes of different clusters were analyzed by GO and KEGG gene enrichment analysis. Differentially expressed genes (DEGs) were screened from different clusters. Consensus cluster analysis of prognostic DEGs was performed to identify gene subtypes. Patients were then randomly divided into training and validation groups, and MTRG scores were constructed by logistic minimum absolute contraction and selection operator (LASSO) and Cox regression analysis. We assessed changes in clinical outcomes and immune-related factors between different patient groups. The single-cell RNA sequencing (scRNA-seq) dataset for BLCA was collected and analyzed from the Tumor Immune Single-cell Hub (TISCH) database. Biological functions were investigated using a series of experiments including quantitative reverse transcriptase polymerase chain reaction (qRT-PCR), wound healing, transwell, *etc*.

**Results**. Our MTRG score is based on eight genes (HTRA1, GOLT1A, DCBLD2, UGT1A1, FOSL1, DSC2, IGFBP3 and TAC3). Higher scores were characterized by lower cancer stem cell (CSC) indices, as well as higher tumor microenvironment (TME) stromal and immune scores, suggesting that high scores were associated with poorer prognosis. In addition, some drugs such as cisplatin, paclitaxel, doxorubicin, and docetaxel exhibited lower IC50 values in the high MTRG score group. Functional experiments have demonstrated that downregulation of DCBLD2 affects tumor cell migration, but not proliferation.

**Conclusions**. Our study sheds light on the prognostic significance of MTRGs within the TME and their correlation with immune infiltration patterns, ultimately impacting patient survival in BLCA. Notably, our findings highlight DCBLD2 as a promising candidate for targeted therapeutic interventions in the clinical management of BLCA.

Corresponding author
Jun Deng, dengjun2004@163.com

## INTRODUCTION

Bladder cancer (BLCA) is one of the most common urinary malignancies worldwide, with more than 300,000 deaths worldwide each year, and the incidence rate is higher in men than in women (*Bray et al., 2018*). The pathogenesis of BLCA involves multiple factors. Genomic studies have shown that some specific gene mutations are closely related to BLCA, such as TP53, RB1, FGFR3, *etc.* (*Knowles & Hurst, 2015*). In addition, environmental and lifestyle factors such as smoking, occupational exposure, and chemical exposure are also widely believed to be associated with BLCA (*Al-Zalabani et al., 2016*). BLCA can be divided into non-muscle-invasive BLCA (NMIBC) and muscle-invasive BLCA (MIBC). Approximately 80% of BLCA patients are NMIBC, with a 5-year survival rate of more than 85% (*Berdik, 2017*). However, NMIBC often progresses to MIBC and metastatic BLCA due to its invasiveness, metastatic propensity, drug resistance, and high recurrence rate. Therefore, it is important to elucidate the prognostic characteristics and underlying mechanisms of BLCA development (*Schneider, Chevalier & Derré, 2019*).

Plasma membrane tension refers to the force per unit length exerted on the cell membrane cross-section, resulting partly from the tension in the lipid bilayer and partly from the adhesion between the membrane and the cytoskeleton (*Chugh et al., 2017*). Membrane tension controls a plethora of biological processes intimately intertwined with diseases, encompassing cell division, cellular mobility, endocytosis, and exocytosis (*Chronopoulos et al., 2020*; *Wirtz, Konstantopoulos & Searson, 2011*). Studies have compared tumor cells with normal cells to identify differences in membrane tension and function (*Zalba & Ten Hagen, 2017*). For a long time, cell mechanics has been considered to be related to the invasion and metastasis of tumor cells. Research has shown that higher membrane tension can effectively inhibit the migration and invasion of tumor cells (*Gensbittel et al., 2021*; *Gossett et al., 2012*). Disruption and reduction of cell membrane tension have also been demonstrated to be essential features of malignant cells, influencing tumor progression by regulating the rate of glycolysis (*Diz-Muñoz, Fletcher & Weiner, 2013*; *Du et al., 2022*).

Membrane tension-related genes (MTRGs) exhibit distinct characteristics in cancer due to tumor heterogeneity and the corresponding tumor microenvironment (TME). This study seeks to extensively explore the function of MTRGs in the prognosis, TME panorama, and immunotherapy of BLCA. We developed and validated a prognostic model associated with MTRGs, effectively assessing immune cell infiltration and predicting drug sensitivity in BLCA patients. Experiments show that DCBLD 2 is a key gene for bladder cancer migration and invasion, and has potential application value in the treatment of bladder cancer.

## METHODS

### Data acquisition

After consulting relevant literature, we selected 41 MTRGs for analysis (Table S1) (*Diz-Muñoz, Fletcher & Weiner, 2013*; *Simunovic et al., 2019*; *Issa & Noureddine, 2017*; *Ha & Chi, 2012*). RNA sequencing data and clinical annotations for BLCA samples were obtained from The Cancer Genome Atlas (TCGA) (TCGA-BLCA, https://www.cancerimagingarchive.net/collection/tcga-blca/) and Gene Expression Omnibus (GEO) repository (https://www.ncbi.nlm.nih.gov/gds). The TCGA database includes 19 normal samples and 412 tumor samples, whereas the GEO (GSE13507) database contains 165 samples with clinical data (*Kim et al., 2010*). Combining the TCGA-BLCA dataset with the GEO dataset, we eliminated batch effects using the Combat algorithm prior to subsequent analysis. Somatic mutation data were downloaded from TCGA, which contained 407 BLCA samples. Immunohistochemical (IHC) images of proteins taken from Human Protein Atlas (https://www.proteinatlas.org/) (*Uhlén et al., 2015*).

### Differential expression analysis of MTRGs and clinical correlation of molecular subtypes

We examined the variance in expression among 41 MTRGs and explored the association between overall survival (OS) and MTRGs. Using the R "Consensus Cluster Plus" software package, we conducted consensus unsupervised cluster analysis based on the differentially expressed MTRGs. This method strengthens intra-group links while weakening inter-group links. As a result, BLCA patients were classified into different molecular subtypes. To explore the clinical significance of these molecular subtypes, we utilized Kaplan–Meier curves to examine their association with overall survival (OS) and clinicopathological characteristics. Clinicopathological features included age, sex, grade, and TNM stage.

### Gene set enrichment analysis and immune profiling analysis

To further investigate the pathways and biological processes of MTRGs molecular subtypes, we conducted a series of enrichment analyses, including GSVA and GSEA analyses. Gene set required for enrichment analysis obtained from MSigDB (https://www.gsea-msigdb.org/gsea/msigdb/) (*Liberzon et al., 2011*). Additionally, we employed the R package "CIBERSORT" to calculate the differences in tumor immune cells (TICs) infiltration between different molecular subtypes (*Newman et al., 2015*).

### Gene subtypes and prognostic model construction

Differentially expressed genes (DEGs) between molecular subtypes based on MTRGs were identified using the R package "limma" (*Ritchie et al., 2015*). A significance level of $P < 0.05$ and $|logFC|>1.5$ was considered meaningful. Following that, we conducted univariate Cox regression analysis to identify DEGs linked to prognosis. Based on consensus cluster analysis of the expression of prognostic DEGs, patients were stratified into different gene subtypes. We further investigated the clinical significance of these gene subtypes and differences in MTRGs expression. The TCGA-BLCA and GSE13507 datasets were divided into training and test sets at a ratio of 1:1 using the R package "caret". MTRG risk scores

were constructed using logistic minimum absolute contraction and selection operator (LASSO) and COX regression in the training set and validated in the test and merged sets. The MTRG risk score was calculated as the sum of the product of the risk coefficient (coef) and gene expression (Exp) for each prognostic model gene, where coef represents the risk coefficient and Exp represents the expression of the prognostic model gene. We visualized risk scores and prognostic outcomes using mulberry graph in relation to different molecular or gene subtypes. The training, test, and merged sets were all stratified into high-risk and low-risk groups based on median risk scores. Survival analyses, heat maps, and receiver operating characteristic (ROC) curves were generated for each group. Additionally, using the R package "rms" and significant clinicopathological features ($p < 0.05$), we developed a nomogram incorporating age, sex, TMN stage, and MTRGs risk score. This nomogram was used to estimate short-, medium-, and long-term overall survival in patients with BLCA. Calibration plot of the prognostic model was generated to verify the accuracy of the model.

## Tumor microenvironment (TME), tumor mutation burden (TMB) and drug sensitivity analysis

We assessed the expression of MTRGs in high- and low-risk groups and calculated differences in the TME across risk groups. Correlation analysis further revealed the relationship between TICs and the eight scoring genes. Mutation frequencies in different risk groups were assessed using the R package "maftools". To investigate differences in drug susceptibility between risk groups, we calculated the half-inhibitory concentration (IC50) values for common therapeutic drugs using the R package "pRRophetic". The method uses drug sensitivity data of Cancer Cell Line Encyclopedia (CCLE) in combination with gene expression data of BLCA samples to estimate 50% inhibitory concentration based on prediction model.

## Immunohistochemistry (IHC) and single cell analysis

Two genes (HTRA1 and DCBLD2, $p < 0.05$) were identified as being positively associated with prognosis by analyzing the association between eight prognostic score genes and overall survival (OS). Immunohistochemical (IHC) images from the Human Protein Atlas database (HPA, https://www.proteinatlas.org/) were used to compare HTRA1 and DCBLD2 expression levels between cancer tissues and adjacent tissues. The expression and distribution of HTRA1 and DCBLD2 in various cell clusters were assessed in the BLCA-GSE130001 single-cell dataset using the Tumor Immune Single-cell Hub (TISCH) database (http://tisch.comp-genomics.org/).

## Verification of DCBLD2 expression in BCLA tissue samples

On one hand, immunohistochemistry showed that HTRA1 was highly expressed in BLCA tissues, although not significantly. On the other hand, considering relevant literature in recent years, we ultimately selected DCBLD2 for further validation. Ten pairs of tumor and adjacent tissues were obtained from BLCA patients who recently underwent radical surgery at the First Affiliated Hospital of Nanchang University. All participants provided written informed consent. Total RNA extraction was performed using TRIzol reagent,

followed by cDNA synthesis using the Takara PrimeScript RT kit. Real-time quantitative PCR was conducted using SYBR Green (Roche, Basel, Switzerland) reagent. Relative gene expression levels were determined using the $2^{-\Delta\Delta Ct}$ method, with β-actin as the endogenous reference. The primer sequences used were as follows:

DCBLD2_F: CCTGCAAAAGCAGTGGACCATG.
DCBLD2_R: CTCCTACCAGTGGCTGAGCATA.

## Cellular cultivation and transfection

Human BLCA cell lines BIU and T24 were cultured in RPMI-1640 and DMEM media, respectively. All cells were incubated in media supplemented with 10% fetal bovine serum (FBS) and 1% penicillin-streptomycin in a humidified incubator at 37 °C and 5% CO2. These cell lines were obtained from Procell Life Science & Technology Co., Ltd (Wuhan, China) and authenticated by the cell bank of the Chinese Academy of Sciences Type Culture Collection Center (Shanghai, China). siRNA targeting DCBLD2 and controls were synthesized by GenePharma, Ltd (Shanghai, China). Cells were seeded into 6-well plates at 50%–60% density per the manufacturer's instructions, and siRNA was transiently transfected using Lipofectamine 2000 (Invitrogen). The siRNA sequences used were as follows:

DCBLD2 siRNA-1 (5′-GUUCCUCCUGCUCUUACUUTT-3′)
DCBLD2 siRNA-2 (5′-GGCCUCAUACUCUGUUAUATT-3′)

## Wound healing and transwell assay

T24 and BIU cells were seeded into six-well plates at 50% density and allowed to grow to near confluence after transfection with siRNA. A cross was scratched onto the dish using a 200 µl pipette tip, and the debris was washed away with PBS. After changing the medium to serum-free medium, images were captured at 0 h under a microscope. The six-well plate was then placed in the incubator for an additional 24 h, and images were taken again after 24 h. The wound healing rate was quantified using ImageJ software.

For the Transwell assay, 40,000 transfected T24 and BIU cells were mixed with 200-µl of serum-free medium and evenly inoculated into the upper chamber. The lower chamber contained 800 µl of medium containing 20% FBS. After incubation for 24 h, cells were fixed with 4% paraformaldehyde and stained with 1% crystal violet. Cells on the lower surface of the chamber were photographed under a light microscope and counted to measure the extent of cell migration.

## Plate cloning and EDU experiments

T24 and BIU cells transfected with siRNAs targeting DCBLD2 and controls were incubated for 24 h, seeded at 1,000 per well into 6-well plates, and cultured in medium containing 10% FBS. After 14 days, they were removed and fixed with 4% paraformaldehyde for 30 min and stained with 1% crystal violet for 30 min. Transfected cells were inoculated into 96-well plates at 5,000 cells per well and incubated for 24 h, then the medium was replaced with EDU reagent and incubated for another 2 h. Subsequent staining was then performed according to the instructions provided in the Edu assay kit (C10310-1; RiboBio), and images were taken under a fluorescence microscope after all staining.

## Statistical analysis

All bioinformatics results were analyzed using R 4.3.2 software, and experimental data were visualized using GraphPad Prism 9.5.1. Student $t$-test was used to compare means between groups, while one-way/two-way ANOVA was used to compare means between groups. For multiple hypothesis testing, the Benjamini–Hochberg (BH) method was used to adjust the $p$-value to control for false discovery rate (FDR). Adjusted $p$-values <0.05 were considered statistically significant.

## Ethics approval and consent to participate

All organizations included in this study obtained ethical approval from the Ethics Committee of the First Affiliated Hospital of Nanchang University (Approval ID: (2024) CDYFYYLK (08-040)), and patient participation was contingent upon informed consent.

# RESULTS

## Differentially expressed MTRGs and molecular subtypes

To determine changes in the expression of MTRGs in BLCA patients, we analyzed the differences in the expression of 41 MTRGs in the TCGA-BLCA dataset. Among them, 24 MTRGs showed different expression levels between tumor and normal tissues ($p < 0.05$, Fig. 1A). The TCGA-BLCA dataset was merged with the GSE13507 dataset, and the Combat algorithm was utilized to eliminate batch effects. Subsequent analyses were conducted on the merged dataset. Survival analysis revealed that 23 MTRGs were significantly associated with overall survival (OS) in BLCA patients (Figs. S1 and S2). Following this, one-way COX regression analysis indicated that 9 MTRGs were associated with survival in BLCA patients (Table S2). Crossover analysis of survival and univariate Cox regression showed that all 9 genes were significantly associated with prognosis in BLCA patients. Next, a plasma membrane tension network was performed to fully demonstrate the association between MTRGs and their prognostic value in BLCA patients. The thickness of the link between two genes represents the strength of the correlation. The results show that there are close and universal interactions between MTRGs.

To further verify the association of MTRGs with the disease, we employed a consensus clustering algorithm and found that $K = 2$ was the optimal choice (high within-group correlation, low between-group correlation, Figs. 1C–1D). Consequently, BLCA patients were classified into two molecular subtypes, with patients in subtype B showing a better prognosis than those in subtype A ($p < 0.05$, Fig. 1E). Principal component analysis (PCA) unveiled substantial disparities between subtypes A and B (Fig. S3A). Additionally, we analyzed the association of MTRGs with clinicopathological characteristics according to different subtypes, including gender, age, TNM stage, grade, and dataset origin (Fig. 2A, clustering details: Rows and columns were hierarchically clustered using the Euclidean distance and complete linkage method.).

## Association between TME and molecular subtypes

To explore the characteristics of different subtypes of the TME, we conducted enrichment analyses using KEGG GSVA and GSEA methods. KEGG GSVA analysis revealed that

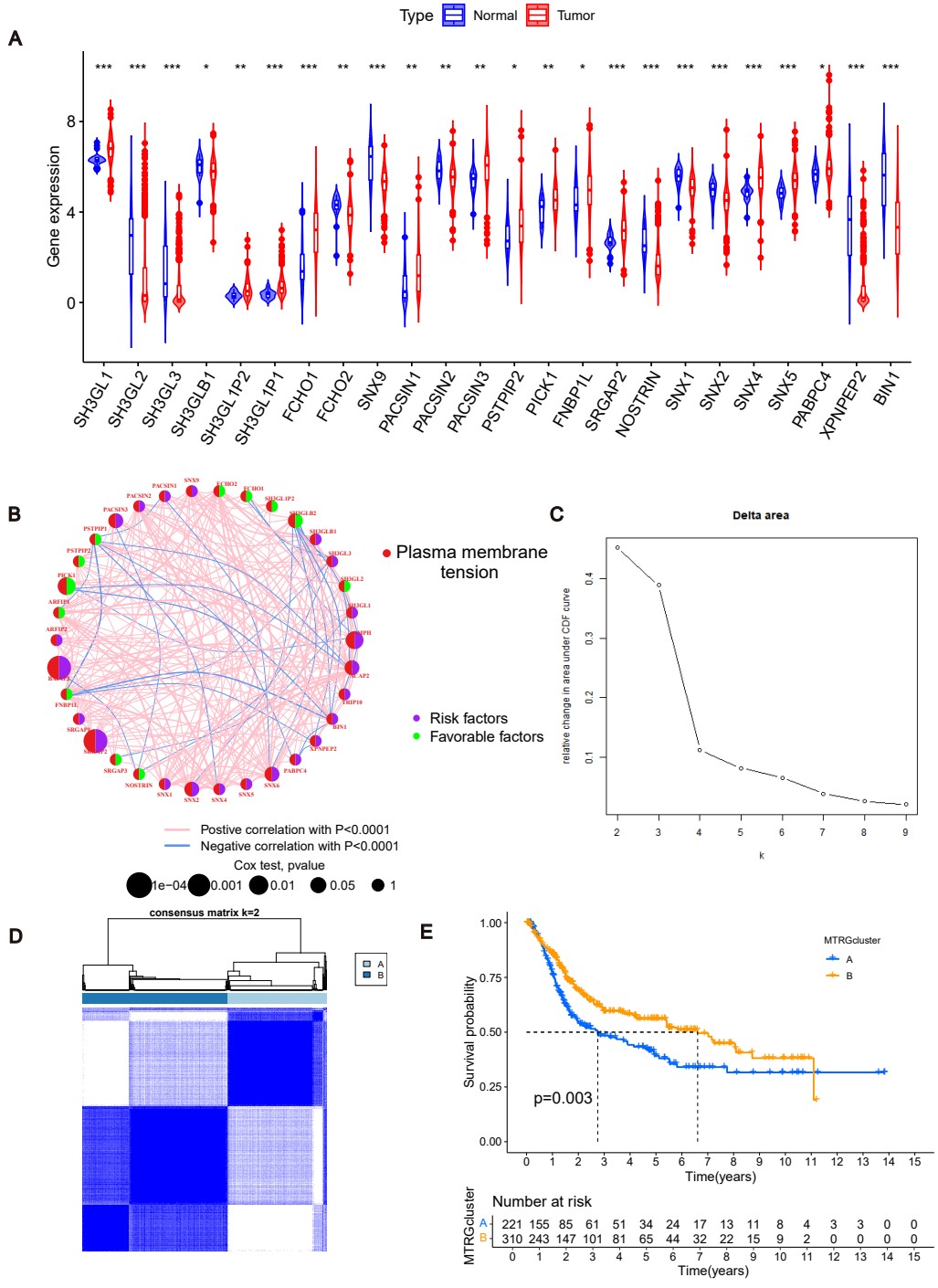

**Figure 1   Differentially expressed MTRGs and molecular subtypes.** (A) Expression level of MTRGs between tumor and normal tissues. (B) Network diagram showing correlation between MTRGs. (C–D) The consensus clustering in BLCA samples with $k = 2$. (E) Kaplan–Meier (KM) survival analysis of two subgroups.

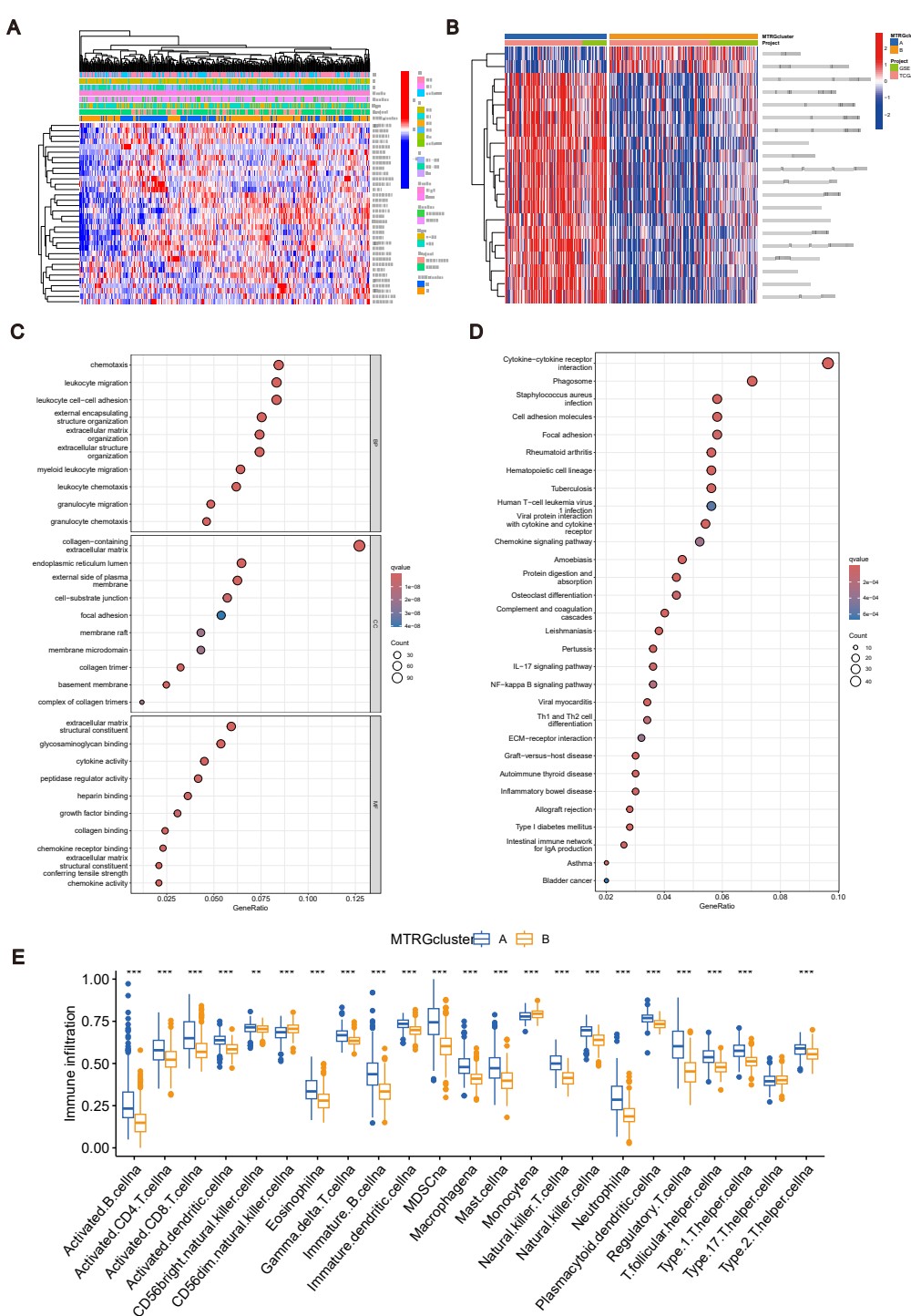

**Figure 2 Association between TME and molecular subtypes.** (A) Correlation between MTRGs and clinicopathological features. (B) KEGG GSVA enrichment assay. (C) Difference in expression of immune cells between subtypes A and B. (D) GO GEVA enrichment analysis. (E) KEGG GEVA enrichment analysis.

subtype B was significantly enriched in peroxidase and glycerolphospholipid metabolic pathways (Fig. 2B). Conversely, subtype A exhibited significant enrichment in multiple pathways, including natural killer cell-mediated cytotoxicity, JAK-STAT signaling, T cell receptor signaling, MAPK signaling, actin cytoskeletal regulation, and focal adhesion (Fig. 2B). Additionally, employing the R package "CIBERSORT", we assessed differences in the expression of immune cells between different subtypes. Our analysis indicated notable infiltration differences between subtype A and B, with Activated B cells, Activated CD4 T cells, macrophages, and Regulatory T cells showing higher expression levels in subtype A compared to subtype B (Fig. 2C).

### Identification of MTRGs associated gene subtypes

Using the R "limma" software package, we identified 952 DEGs between subtypes A and B (Table S3) and performed GEVA enrichment analysis of the DEGs. GO GEVA enrichment analysis showed that the pathways were mainly concentrated in extracellular matrix components, leukocyte chemotaxis, endoplasmic reticulum lumen, cell adhesion and collagen binding. The KEGG GEVA enrichment assay revealed more cytokine receptors, chemokines, and cell adhesion-related pathways. In conclusion, we hypothesize that DEGs play an important role in the regulation of cell structure and adhesion. Further univariate COX regression analysis of DEGs was used to determine which DEGs were associated with prognosis. Finally, we screened out 447 DEGs (Table S4). Subsequently, we performed consensus cluster analysis of 447 prognostic relevant DEGs (Fig. 3A, Fig. S3B). Based on the results of the analysis, we reclassified the patients into two gene subtypes (gene subtype A, gene subtype B). Similarly, OS differed between gene subtypes, with patients with gene subtypes B having a better prognosis than those with gene subtypes A ($p < 0.05$, Fig. 3B). We reassessed the association of the three (gene subtype, clinicopathological features, and molecular subtype) as shown in the heat map (Fig. 3C). We also assessed the differences in MTRGs expression across gene subtypes and showed that a total of 29 MTRGs showed differences across gene subtypes ($p < 0.05$, Fig. 3D).

### Identification of MTRG scoring system and nomogram

The combined TCGA-BLCA and GSE13507 dataset ($n = 531$) was split into a training dataset ($n = 266$) and a test dataset ($n = 265$) at a 1:1 ratio using the R package "caret". Subsequently, LASSO and multivariate COX regression (Figs. 3E–3F) were performed on the 447 prognosis-related DEGs in the training set. Consequently, eight risk score genes (HTRA1, GOLT1A, DCBLD2, UGT1A1, FOSL1, DSC2, IGFBP3, and TAC3) were obtained. Based on the risk scores of these eight genes, we constructed the MTRG risk scoring system as follows:

$$\text{MTRG Risk Score} = \sum_{i=1}^{n} Exp_i \beta_i. \tag{1}$$

In the given equation, "Exp" and "$\beta$" denote the adjusted expression value and regression coefficient of gene i, respectively. Utilizing patient clinical data, we constructed a mulberry graph to depict the correlation between molecular subtype , gene subtype , risk group, and final outcome (Fig. 3G).

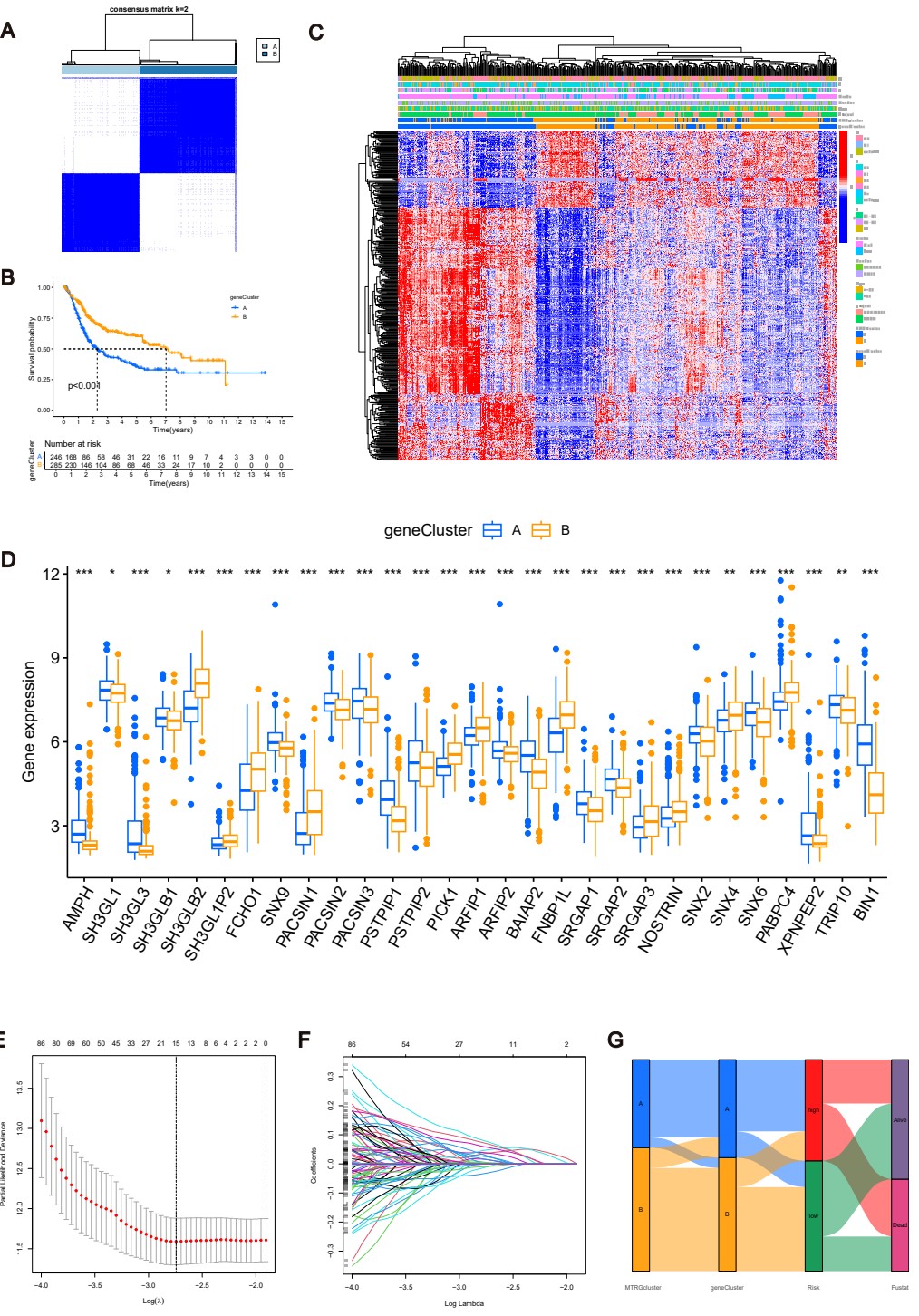

**Figure 3   MTRGs associated gene subtypes.** (A) Genotype consensus clustering of BLCA samples, $k = 2$. (B) KM survival analysis of two genotypes. (C) Correlation of genetic subtypes, clinicopathological features and molecular subtypes. (D) Expression levels of MTRGs in different gene subtypes. (E) Coefficient Plot of 8 Prognostic MTRGs. (F) The least absolute contraction and selection operator (LASSO) analysis identified 8 prognostic genes. (G) Mulberry graph.

Similarly, we calculated risk scores for each BLCA sample for different molecular subtypes and gene subtypes. The risk scores of molecular subtype A and gene subtype A were significantly higher than those of molecular subtype B and gene subtype B (Fig. S3C), respectively, consistent with the previous Kaplan–Meier (KM) curve (Figs. 1E and 3B). Examining the expression of MTRGs between the high-score and low-score groups, we noted that 11 MTRGs were upregulated in the low-score group, while 18 MTRGs were upregulated in the high-score group (Fig. 4A).

To validate the accuracy of the scoring system, we employed K–M curves, ROC curves, heat maps, risk score distributions, and survival scatter plots. Validation was performed separately in the merged set, training set, and test set. The Kaplan–Meier curves showed that patients with lower scores showed better OS (Figs. 4B–4D). ROC curves indicated that the MTRG risk score predicted 1-, 3-, and 5-year survival with high sensitivity and specificity (Figs. 4E–4G). Heatmaps depict differences in scoring genes among different scoring groups, with HTRA1 and DCBLD2 significantly expressed in the high-risk group (Figs. 4H–4J). Scatter plots of risk score distribution and survival showed that higher risk scores corresponded to shorter survival times (Fig. S3D).

Consistent results across the merged set, training set, and test set validated the scoring system's high accuracy. By combining relevant clinical characteristics and the risk scoring system, we constructed a nomogram (Fig. 4K) to assess OS at 1, 3, and 5 years in BLCA patients. Through calibration curve analysis (Fig. 4L), we confirmed the predictive ability of our constructed nomogram credible.

## Correlation analysis of MTRG risk score with TME and TMB

To further explore the relationship between the MTRG risk score system and TME, We identified eight risk score genes associated with TIC abundance. As shown in Fig. 5A, eight genes are strongly associated with most TICs, especially T cells regulatory (Tregs). Immunocyte correlation analysis showed that macrophages M0, activated mast cells and neutrophils were positively correlated with risk score, while monocytes, T cells CD8 and regulatory T cells (Treg) were negatively correlated with risk score ($p < 0.05$, Fig. S3E). The high score group showed higher levels of StromalScore, ImmuneScore, and ESTIMATEScore than the low score group ($p$ 0.001, Fig. 5B). Hence, we speculate that the MTRG risk score could be linked to the TME of BLCA. Some studies have shown that tumor stem cells (CSC) are the fundamental factors of tumorigenesis, drug resistance, recurrence and metastasis, and also an important reason for tumor treatment failure (*Patel, Oh & Galsky, 2020*; *Dobruch et al., 2016*). Therefore, we further explored the correlation between the MTRG scoring system and the CSC index. The results showed a negative correlation between score and CSC index ($R = -0.13$, $P < 0.001$, Fig. 5C). Mutation frequencies in different risk groups were analyzed by the R "maftools" program. The analysis revealed that the mutation frequencies of TP53, KMT2D, MUC16, SYNE1, STAG2, ELF3, RB1, RYR2, KMT2C, MACF1, OBSCN, and CSMD3 were elevated in the high-score group (Figs. 5D–5E). However, further analysis of TMB reflecting the number of tumor mutations found no clinically significant difference in TMB between the high and low scoring groups ($R = 0.039$, $p = 0.46$, Figs. S4B–S4C).

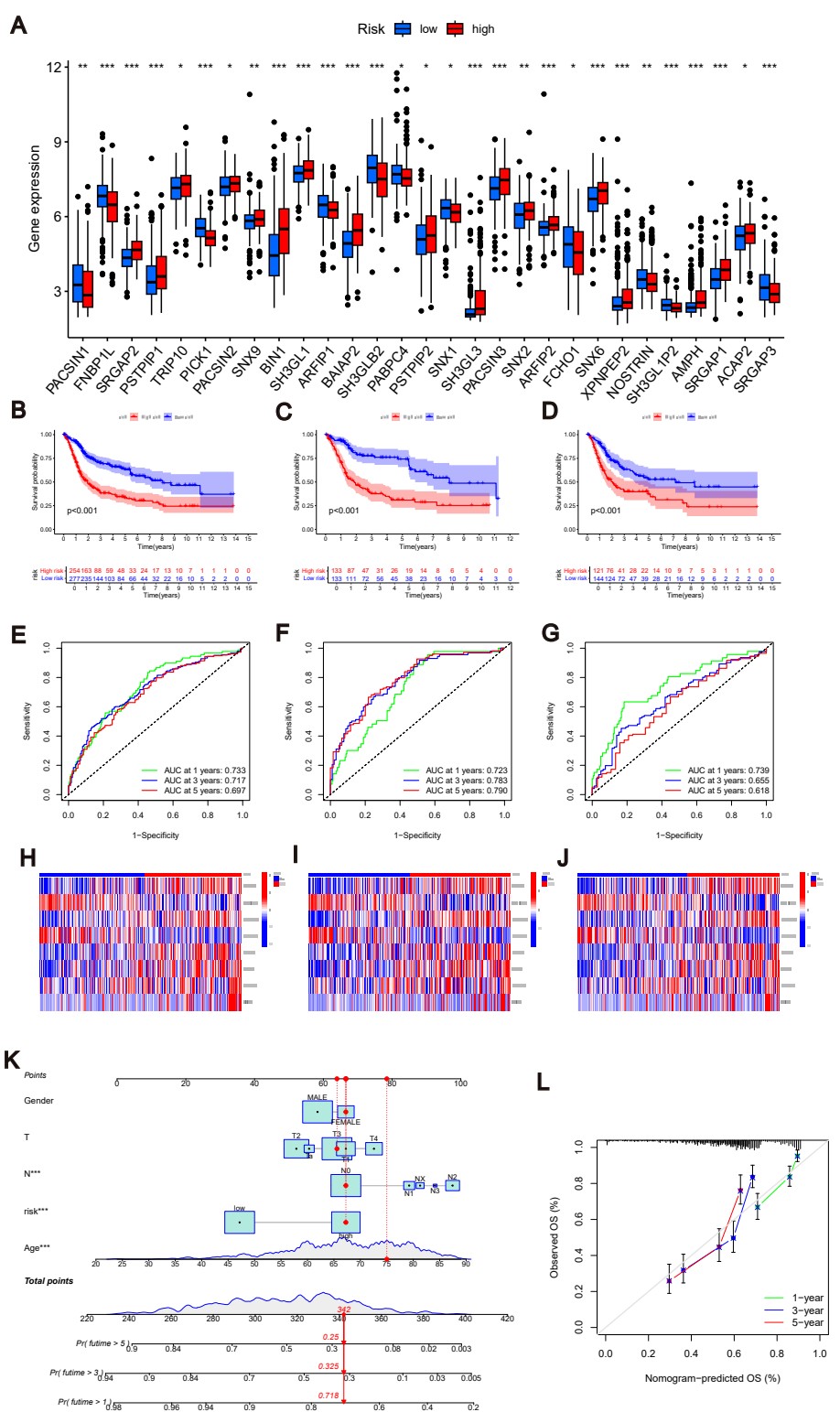

**Figure 4** Construction of nomograms. (A) Expression levels of MTRGs in different scoring groups. (continued on next page...)

**Figure 4 (…continued)**
(B–D) KM curves corresponding to different scoring groups in the combined set, training set and test set are shown respectively. (E–G) ROC curves corresponding to different scoring groups in the pooled set, training set and test set. (H–J) The expression of eight risk score genes in different score groups of pooled set, training set and test set were shown respectively. (K) Nomograms used to assess the 1-, 3-, and 5-year OS in patients with BLCA. (L) Calibration curve for nomograms.

## Identification of drug susceptibility and scoring gene expression levels

Using the R "pRRophetic" software package, we computed IC50 values for several common drugs utilized in BLCA treatment. A lower IC50 value indicates that the sample is more sensitive to a particular drug. We combined this information with the results of the high-risk and low-risk groups to assess the potential therapeutic effect of different drugs in each risk group. The findings indicated that certain drugs such as cisplatin, paclitaxel, doxorubicin, and docetaxel exhibited lower IC50 values in the high-scoring group (Figs. 5F–5I), suggesting lower drug resistance in the high-risk group. To delve deeper into the biological function of risk score genes, we individually analyzed the association of the eight score genes with OS. The analysis revealed a negative correlation between DCBLD2 and HTRA1 expression and prognosis ($p < 0.05$, Figs. 5J–5K, Fig. S4A). The remaining six genes showed no clinically significant impact on OS and were therefore excluded from subsequent analyses. BLCA samples were queried in the Human Protein Profile (HPA) database to access immunohistochemical images of DCBLD2 and HTRA1 (Fig. 6A). The results indicated that DCBLD2 exhibited moderate to high expression in BLCA compared to normal bladder tissue, whereas HTRA1 showed lower expression levels in BLCA relative to DCBLD2. To further validate the distinct expression of DCBLD2 and HTRA1 at the single-cell level, we utilized a single-cell dataset (BLCA-GSE130001) from the TISCH database to explore gene expression differences among different subtypes. The analysis revealed that DCBLD2 was predominantly expressed in fibroblasts and myofibroblasts, while HTRA1 was primarily upregulated in endothelial cells and fibroblasts (Figs. 6B–6D). Figure 6E shows the composition of various cell types, and the correlation between DCBLD2 and HTRA1 is shown in Fig. S3D.

## Cell identification *in vitro*

PCR analysis of DCBLD2 using 10 tissue pairs confirmed its significant expression in BLCA tissues (Fig. 7A). Also, using PCR, we verified the knockdown efficiency of Si1-DCBLD2 and Si2-DCBLD2 in T24 and BIU cell lines (Fig. 7B). Through wound healing experiments and Transwell assays, we found that decreased DCBLD2 levels was associated with reduced migration and invasion of T24 and BIU cells (Figs. 7C–7D). However, colony formation and EDU experiments showed contrasting results. The findings indicated that decreased DCBLD2 expression did not affect the proliferation of T24 and BIU cells (Figs. 7E–7F). Therefore, based on the results of *in vitro* cell experiments, we can infer that DCBLD2 is pivotal in facilitating BLCA cell migration and invasion but does not notably impact proliferation.

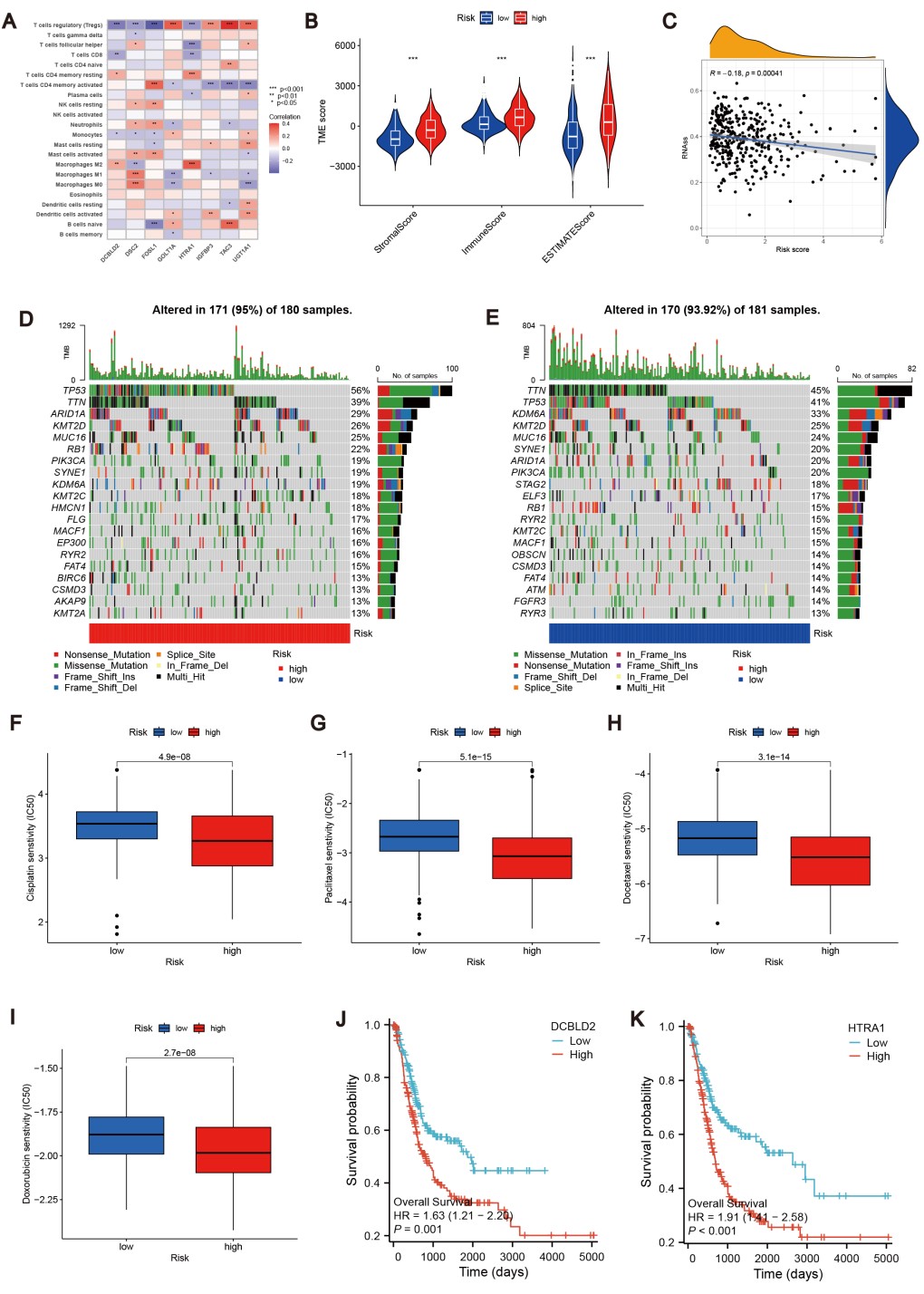

**Figure 5** **Correlation analysis of MTRG risk score with TME, TMB and drug susceptibility.** (A) Correlation between 8 risk score genes and TIC abundance. (B) Expression levels of StromalScore, ImmuneScore and ESTIMATEScore in different scoring groups. (C) Relationship between MTRG scoring system and CSC index. 

**Figure 5 (…continued)**
(D–E) Waterfall plot of somatic mutation characteristics in different scoring groups. (F–I) IC50 values of common drugs (cisplatin, paclitaxel, doxorubicin and docetaxel) in different scoring groups. (J) Correlation between expression of DCBLD2 and prognosis. (K) Correlation between expression of HTRA1 and prognosis.

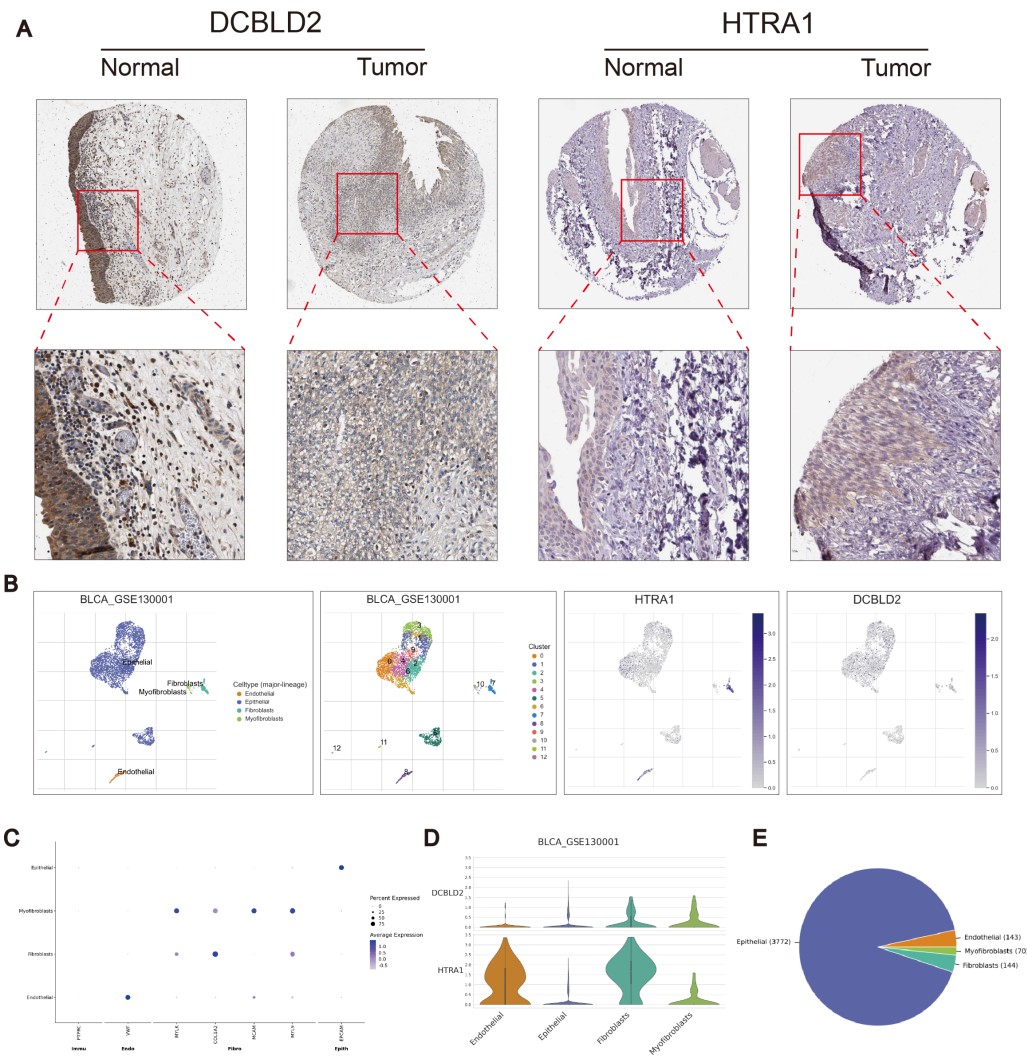

**Figure 6** **Immunohistochemical differences and single cell analysis.** (A) Immunohistochemical differences between DCBLD2 and HTRA1 in tumor and normal tissues. (B) Expression of DCBLD2 and HTRA1 in bladder cancer single-cell dataset (GSE130001). (C) Marker gene expression. (D) Expression of DCBLD 2 and HTRA 1 in different cell types. (E) Statistics of different cell types.

# DISCUSSION

Bladder cancer, as the most common malignant tumor worldwide, has maintained persistent incidence and mortality rates for a long time (*Patel, Oh & Galsky, 2020*). Some studies have shown that family inheritance, smoking, and occupational exposure to

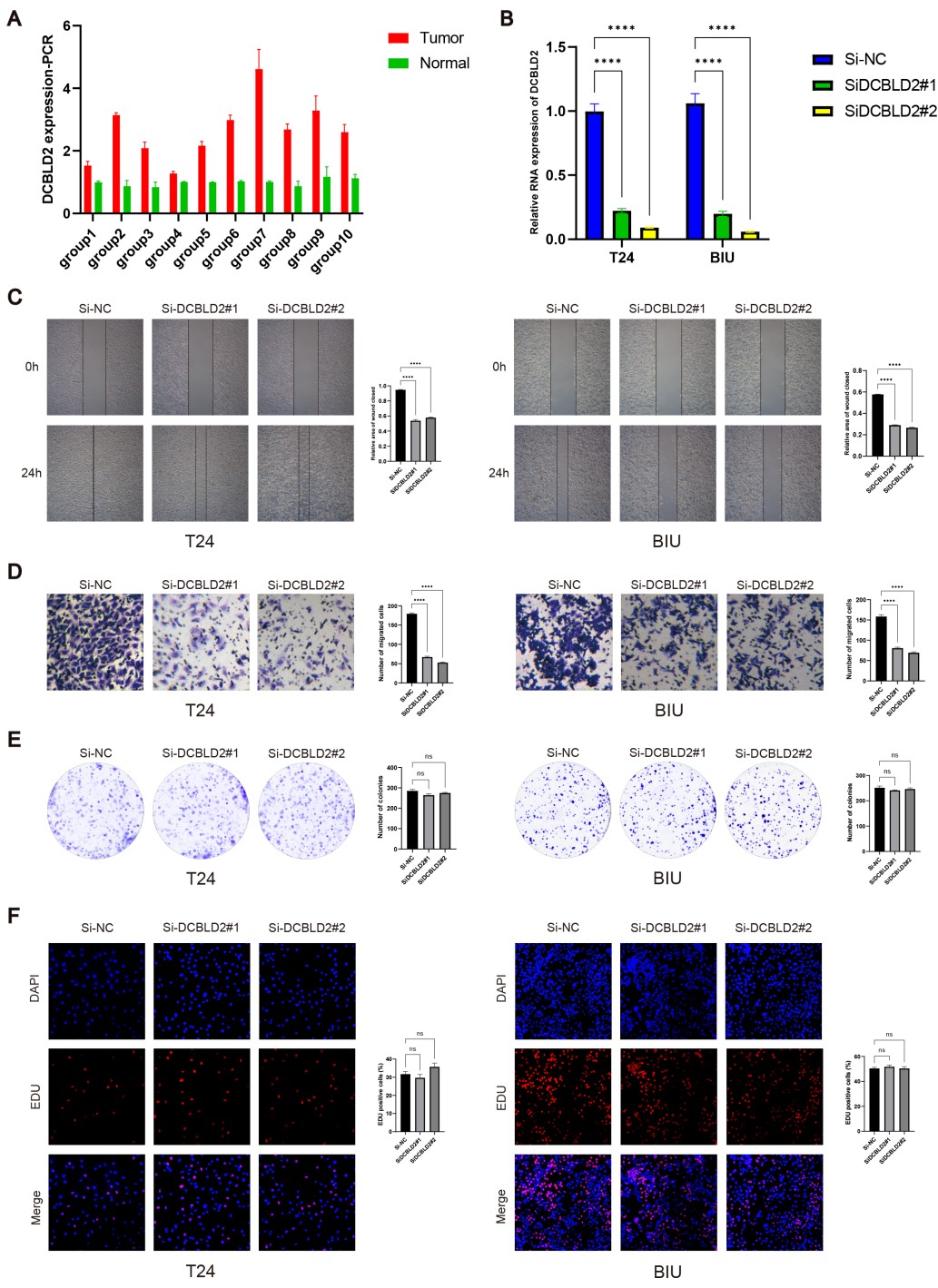

**Figure 7 Functional validation of DCBLD2 *in vitro*.** (A) PCR was used to detect DCBLD2 expression in 10 pairs of BLCA tissues. (B) Knockdown efficiency of small interfering RNA (SiRNA) in T24 and BIU cell lines. (C) Wound healing experiments. (D) The Transwell experiment. (E) Colony formation experiments. (F) EDU experiment.

chemicals are closely related to the occurrence of bladder cancer (*Dobruch et al., 2016*). With the development of molecular biology and genomics, more molecular markers have been found to be closely related to the pathogenesis and therapeutic response of bladder cancer. Therefore, finding new molecular biomarkers for the prognosis of BLCA patients is crucial.

In recent years, many studies have demonstrated that plasma membrane tension proteins related to cell mechanics are associated with tumor cell migration and invasion (*Vogel & Sheetz, 2006*; *Wang, Butler & Ingber, 1993*). The ERM protein is a molecule that maintains membrane-actin adhesion on cell membranes. Disruption of the ERM protein results in decreased cell membrane tension. In metastatic cells, tumor invasion and metastasis are inhibited by manipulating membrane-to-cortex attachment to increase plasma membrane tension and by impairing BAR membrane-mediated mechanical signaling (*Tsujita et al., 2021*).

In this study, 41 MTRGs were identified after reviewing previously published literature. Using transcriptome and clinical information from BLCA samples obtained from TCGA and GEO databases, we assessed the differential expression of MTRGs in normal and tumor tissues. Analysis of these differentially expressed genes for correlation and clinical prognostic value revealed that most of the genes were closely associated with prognosis. BLCA samples were classified into two molecular subtypes (A, B) based on MTRGs, with the B molecular subtype having a higher survival rate than the A molecular subtype. Therefore, we conclude that MTRGs may be potential targets for BLCA therapy.

Similarly, different molecular subtypes were significantly associated with clinical characteristics such as gender, age, TNM stage, and OS. GSVA enrichment analysis showed that subtype A was significantly enriched in signal transduction-related pathways, while subtype B was mainly enriched in metabolism-related pathways. It is well known that immunotherapy plays an important role in the treatment of bladder cancer. Therefore, in the immune-related analysis, we used the R "CIBERSOFT" software package to map the differences in immune cell infiltration among the different molecular subtypes. We found that most of the immune cells had higher infiltration in subtype A.

Based on molecular subtypes A and B, we obtained 952 DEGs. Further GO and KEGG analysis of these differential genes showed that these DEGs play an important role in the regulation of cell structure and adhesion. Based on univariate COX regression analysis of 952 DEGs, we obtained 447 DEGs associated with prognosis. Based on the expression and consensus cluster analysis of the DEGs associated with prognosis, we classify patients into different gene subtypes. The expression of MTRGs varied significantly among different gene subtypes. Clinical characteristics such as gender, age, TNM stage, and OS were also significantly different between different gene subtypes.

After the data set of TCGA and GEO was divided into a training set and a test set in a ratio of 1:1 by the R "caret" software package, LASSO and multivariate COX regression analysis were performed on 447 prognostic DEGs in the training set. Finally, we obtained eight genes associated with risk scores (HTRA1, GOLT1A, DCBLD2, UGT1A1, FOSL1, DSC2, IGFBP3, and TAC3).

HTRA1 is a serine protease that plays important roles in cell proliferation, migration, and apoptosis. Studies have shown that HTRA1 expression is associated with immune cell infiltration and survival in breast cancer and promotes the transdifferentiation of normal fibroblasts into cancer-associated fibroblasts by activating the NF-$\kappa$B/bFGF signaling pathway in gastric cancer (*Wu et al., 2019*). GOLT1A is a Golgi transporter protein. Circ_PDZD8 has been shown to alleviate lidocaine's inhibitory effect on malignant tumor cells by regulating the miR-516b-5p/GOLT1A axis (*Zi, Chen & Ruan, 2022*). DCBLD2 is a key protein that activates the AKT pathway and affects the emergence and development of many diseases (*Kikuta et al., 2017*; *He et al., 2020*; *Coppo et al., 2021*). For example, DCBLD2 can promote tumor metastasis by stimulating EMT (*He et al., 2020*; *Chen et al., 2021*). The UGT 1A1 gene is an important component in the metabolism and nucleation of glucose of certain drugs, including irinotecan and govitecan. Thus, various UGT 1A1 polymorphisms that result in decreased UGT 1A1 enzyme function may lead to an increased risk of treatment-related side effects (*Nelson et al., 2021*). In bladder cancer, HOXA10 may accelerate metastasis by regulating FOSL1 expression (*Cui et al., 2020*). In gastric cancer, DSC 2 inhibits tumor growth by inhibiting the nuclear translocation of $\gamma$-catenin and the PTEN/PI3K/AKT signaling pathway (*Sun et al., 2023*). In prostate cancer cells, DSC2 expression was increased. Inhibition of DSC2 promotes proliferation, colony formation, migration, and invasion of LNCaP cells and PC-3 cells, and inhibits apoptosis of LNCaP cells and PC-3 cells, which provides a basis for the treatment of prostate cancer (*Jiang & Wu, 2020*). CDK12 inhibits insulin-like growth factor binding protein 3 (IGFBP 3) in regulating angiogenesis in advanced prostate cancer (*Zhong et al., 2024*). Tachykinin 3 (TAC3) has been shown to affect gingival oral squamous cell carcinoma cells possibly through tachykinin receptor 3 (TACR3) in bone matrix (*Obata et al., 2016*).

The accuracy of the risk scoring system was verified using KM curves, ROC curves, heat maps, risk score distributions, and survival plots, and was verified in the merged set and test set, respectively. Combining relevant clinical characteristics and risk scoring systems, we constructed a nomogram to assess 1-, 3-, and 5-year OS in BLCA patients. The calibration curve further verifies that the predictive power of the nomograms we construct is credible. Based on predictive models, we extrapolated that the higher the risk score, the lower the survival rate for patients. Subsequently, we further analyzed the association between risk scores and TME. The results showed that the high-scoring group exhibited higher levels of StromalScore, ImmuneScore, and ESTIMATEScore than the low-scoring group. Eight of these genes were strongly associated with most TICs, especially T cells regulatory (Tregs). CCL2 inhibition in mouse models of bladder cancer *in situ* has been shown to inhibit tumor growth, reduce MDSCs and TPCs, and promote tumor immunosuppression (*Liang et al., 2023*). Therefore, we hypothesized that the MTRG risk score was closely related to the TME of BLCA. A negative correlation between the MTRG scoring system and the CSC index indicates that higher risk scores have lower cell stemness. However, further analysis of TMB reflecting the number of tumor mutations revealed that MTRGs were not statistically significant with TMB, which requires further exploration in the future. By drug sensitivity analysis, we found that cisplatin, paclitaxel, doxorubicin, and docetaxel had IC50 values lower in the high-scoring group and higher in the low-scoring group.

Therefore, the high score group has higher sensitivity to these drugs, which also provides important reference for clinical use. Eight risk score genes were analyzed separately for their association with OS, and we further screened two prognostic genes (HTRA1 and DCBLD2). HPA and TISCH databases were used to further analyze the expression differences of the two genes in normal tissues and tumor tissues. The results showed that DCBLD2 and HTRA1 were highly expressed in tumor tissues, and DCBLD2 was mainly expressed in fibroblasts and myofibroblasts, HTRA1 was mainly upregulated in endothelial cells and fibroblasts. Looking at the published literature in recent years, we found that the exploration of DCBLD2 in BLCA is still limited. Therefore, we aim to elucidate the specific role of DCBLD2 in BLCA. Our results show that DCBLD2 mRNA is upregulated in BLCA tissues, and knockdown experiments indicate that BLCA cell migration and invasion are significantly inhibited, while cell proliferation is not inhibited. Our study revealed significant differences in TME between scoring groups, and this difference was also evident in immunotherapy response. This is consistent with previous studies showing that TME plays an important predictive role in immunotherapy outcomes.

However, our study still has some limitations. First of all, this is a retrospective study based on a large public database, and further validation through large-scale clinical studies is necessary. Second, we lacked further studies to investigate the mechanisms by which DCBLD2 affects tumor cell migration and invasion. In addition, retrospective and indirect predictions of immunotherapy response highlight the need for prospective trials involving larger patient cohorts to enhance the reliability of scoring systems.

## CONCLUSION

In this study, we explored the characteristics of MTRGs in BLCA and developed a prognostic model related to plasma membrane tension. The model showed excellent performance in predicting prognosis and immunotherapy response, and was able to evaluate the sensitivity of patients to chemotherapy drugs, providing a reference basis for individualized treatment. In addition, we revealed the specific role of DCBLD2 in the tumor microenvironment and its critical impact on BLCA progression by regulating cell migration and invasion mechanisms, highlighting its potential as a therapeutic target for BLCA. These findings provide new theoretical basis and application prospect for optimizing treatment strategy of BLCA patients.

## ACKNOWLEDGEMENTS

The authors express their gratitude for the collaborative efforts of the TCGA and GEO databases.

### Funding

This study was supported by Science and Technology Program Fund of Health Committee of Jiangxi Province (202210361) and Clinical Research Cultivation Program of First Clinical

Hospital of Nanchang University (YFYLCYJPY202313). The funders had no role in study design, data collection and analysis, decision to publish, or preparation of the manuscript.

## Grant Disclosures

The following grant information was disclosed by the authors:

Science and Technology Program Fund of Health Committee of Jiangxi Province: 202210361.

Clinical Research Cultivation Program of First Clinical Hospital of Nanchang University: YFYLCYJPY202313.

## Competing Interests

The authors declare there are no competing interests.

## Author Contributions

- Zhipeng Wang performed the experiments, prepared figures and/or tables, and approved the final draft.
- Sheng Li performed the experiments, authored or reviewed drafts of the article, and approved the final draft.
- Fuchun Zheng performed the experiments, prepared figures and/or tables, and approved the final draft.
- Situ Xiong analyzed the data, authored or reviewed drafts of the article, and approved the final draft.
- Lei Zhang analyzed the data, authored or reviewed drafts of the article, and approved the final draft.
- Liangwei Wan performed the experiments, prepared figures and/or tables, and approved the final draft.
- Chen Wang analyzed the data, prepared figures and/or tables, and approved the final draft.
- Xiaoqiang Liu conceived and designed the experiments, authored or reviewed drafts of the article, and approved the final draft.
- Jun Deng conceived and designed the experiments, authored or reviewed drafts of the article, and approved the final draft.

## Human Ethics

The following information was supplied relating to ethical approvals (*i.e.*, approving body and any reference numbers):

All organizations included in this study obtained ethical approval from the Ethics Committee of the First Affiliated Hospital of Nanchang University (Approval ID: (2024) CDYFYYLK (08-040)), and patient participation was contingent upon informed consent.

## Data Availability

The data is available at TCGA-BLCA and GEO: GSE13507.

## Supplemental Information

Supplemental information for this article can be found online at http://dx.doi.org/10.7717/peerj.18816#supplemental-information.

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
