# Peer review of "Construction and validation of prognosis and treatment outcome models based on plasma membrane tension characteristics in bladder cancer"

_PeerJ, doi:10.7717/peerj.18816_

## Round 0.1 · original submission · Minor Revisions

Please address concerns of the reviewers and amend manuscript accordingly

Reviewer 1 ·

Basic reporting

Title of the Manuscript
Construction and validation of prognosis and treatment
outcome models based on plasma membrane tension
characteristics in bladder cancer
Authors: Zhipeng Wang, Fuchun Zheng, Sheng Li, Situ Xiong, Lei Zhang, Liangwei Wan, Chen Wang,
Xioanqiang, Jun Deng

General Comments
In current manuscript, Wang et al. assessed MTRG scores of a pool of genes using both bioinformatics and experimental approaches.

Experimental design

Experimental design is Ok

Validity of the findings

Minor comments
1. What is the rationale behind study MTRG in BLCA only Higher? Is this specific to BLCA or could be relevant to other cancer types as well.
2. Higher scores were obtained for low CSC, which is surprising as CSC contribute to cancer relapse and sometimes advance cancer stages. Please explain what could be the possible reason for this observation or justify otherwise.
3. How high/low MTRG score correlated (positively/negatively) to susceptibility to multiple drugs. ?
4. Line94: What is Genset?
5. DCBLD2 is already known to associated with poor prognosis, immune regulation and drug sensitivity in BLCA? What could be novel dimension in context to the current research work?
6. Conclusion is week, needs improvement

Reviewer 2 ·

Basic reporting

Clear and unambiguous, professional English used throughout: The English used in the manuscript is satisfactory.

Literature references, sufficient field background/context provided: Yes

Professional article structure, figures, tables. Raw data shared: Data is retrieved from literature, and accession details are provided. However scripts developed for the calculations is not shared.

Self-contained with relevant results to hypotheses: Yes

Experimental design

In this section, I will list the points where the manuscript should be improved:
1. Names of the disease subtypes are confusing: What do the 'molecular', 'genetic', and 'genotype' subtypes refer to, I believe they are the same? The best name seems to be 'molecular subtypes (based on gene expression levels)'.
2. Were effect of the age, sex etc regressed out? I believe they were not, in Discussion it was mentioned that subtypes are enriched for patients from same age range or same sex group which indicates that regression is needed.
3. What is 'pooled' and 'merged' group of data?
4. How was the ROC curve for subgroups made, which data was used, how were TP and TN defined?
5. Formulation for the MTRG risk score was explained but no mathematical equation was provided. It should be provided.
6. In Fig1A, genes from tumor samples seems to have a bimodal distribution, a violin plot would be better than a box plot for this data.
7. In Fig1D-E Clusters were first called 1 and 2, then A and B, the naming should be same in both.
8. Table S2 - Only 9 genes seems to significant, but in text 34 significant genes are mentioned.
9. Which p-value correction method was used for the tests (in all methods used in the manuscript)?
10. Fig2A: This figure would look clearer if genes were clustered hierarchially
11. Supp. Fig.3A - % variations explained by the PCs should be provided.
12. Table S4 - Number of significant genes seems to be 190, not 433 as mentioned in the text.
13. No details on how IC50 were calculated how IC50 should be interpreted is provided.
13. Supplementary tables should be provided in Excel format.
14. PDF files for ethical approvals were corrupt.
15. Fig3A and Fig3C would benefit from hierarchical clustering of the genes.
16. Typo in Lines303-304 should be corrected (figure names)
17. Incomplete sentence in Lines346-347

Validity of the findings

Meaningful replication encouraged where rationale & benefit to literature is clearly stated: The scripts used were not provided, which makes reprıdction of the results impossible:

All underlying data have been provided; they are robust, statistically sound, & controlled: P-value correction methods and scripts used were not provided.

Conclusions are well stated, linked to original research question & limited to supporting results: Yes

Additional comments

I believe some of the methods were redundant in investigation of the profiling of the subgroups. But readers may find it assuring that multiple methods point to the same conclusion.

---

## Round 0.2 · accepted · Accept

The authors have adequately addressed all of the reviewers' comments and made the necessary amendments in the manuscript. Revised version is acceptable now.